# A Critical Review of the Material Characteristics of Additive Manufactured IN718 for High-Temperature Application

**Ching Kiat Yong** [1],*, **Gregory J. Gibbons** [1], **Chow Cher Wong** [2] **and Geoff West** [1]

1 Warwick Manufacturing Group (WMG), University of Warwick, Coventry CV4 7AL, UK; G.J.Gibbons@warwick.ac.uk (G.J.G.); G.West@warwick.ac.uk (G.W.)
2 Advanced Remanufacturing and Technology Centre (ARTC), CleanTech Two, Singapore 637143, Singapore; wongcc@artc.a-star.edu.sg
* Correspondence: ching-kiat.yong@warwick.ac.uk; Tel.: +44-65-82000994

**Abstract:** This paper reviews state of the art additive manufactured (AM) IN718 alloy intended for high-temperature applications. AM processes have been around for decades and have gained traction in the past five years due to the huge economic benefit this brings to manufacturers. It is crucial for the scientific community to look into AM IN718 applicability in order to see a step-change in production. Microstructural studies reveal that the grain structure plays a significant role in determining the fatigue lifespan of the material. Controlling IN718 respective phases such as the Y", δ and Laves phase is seen to be crucial. Literature reviews have shown that the mechanical properties of AM IN718 were very close to its wrought counterpart when treated appropriately. Higher homogenization temperature and longer ageing were recommended to dissolve the damaging phases. Various surface enhancement techniques were examined to find out their compatibility to AM IN718 alloy that is intended for high-temperature application. Laser shock peening (LSP) technology stands out due to the ability to impart low cold work which helps in containing the beneficial compressive residual stress it brings in a high-temperature fatigue environment.

**Keywords:** laser powder bed fusion; Inconel 718; high temperature; material characterisation; laser shock peening

## 1. Introduction

Additive manufacturing (AM) is a promising technology for fabricating a wide range of structures and complex geometries from three-dimensional (3D) model data. The process consists of depositing successive layers of material, one layer on top of another. AM was first developed by Chuck Hull in 1983, who established the process which was later known as stereolithography [1]. Designs are drawn using a computer-aided design (CAD) program which is then translated into model data. A 3D printer takes this data and slices it into several dimensional plans which instruct it where to deposit the layers of material. In 2015, the American Society of Testing and Materials (ASTM) issued a standard for AM technologies that consists of seven main processes [2], which established and defines the terms used in the field.

Alloys used for high-temperature application are highly sought after in the aerospace and nuclear industry due to their high strength and stability at extreme temperatures. Alloys that operate at high temperature are critical for these industries as the efficiency of fuel conversion is closely related to the operating temperature. Generally, these alloys are nickel-, iron- or cobalt-based. Their strength could sometimes become a weakness, as machining these alloys can be very difficult and expensive due to

the natural tendency for work hardening. The shift to AM technology has allowed manufacturers to produce complex geometries such as lattice structures [3,4], where traditional manufacturing such as casting or forging are a lot more time-consuming, or incapable of achieving these geometries.

This review paper aims to provide an overview of the AM of Inconel 718 (IN718), focusing on powder-bed fusion (PBF), which is one of the seven AM technologies. To the best of the authors' knowledge, a robust understanding of the fatigue response for AM IN718 in a high-temperature environment remains elusive. The literature review covers the current research gap and challenges encountered in adopting AM IN718 for commercial use. Microstructural development and mechanical performances of AM IN718 are the focus in this area. Effects such as build orientation and the AM thermal history are purposely left out as reviews in this area have been extensively done by the scientific community. Post-processing methods for AM IN718 are explored, as the scientific community sought out ways to push its usability for high-value applications [5–11].

## 2. Additive Manufacturing (AM)

### 2.1. Benefits of the AM Process

Researchers have made great efforts to understand the process–structure–property–performance relations of AM materials [12]. It is crucial to understand the consequence of each additional process that will impact the material's performance by altering its structure. Figure 1 illustrate a general material design chart with the intent to produce the optimal mechanical properties suited for its specific application. In the upcoming sections, material-specific heat treatment will be brought up frequently and the effect of IN718 material properties on its mechanical properties.

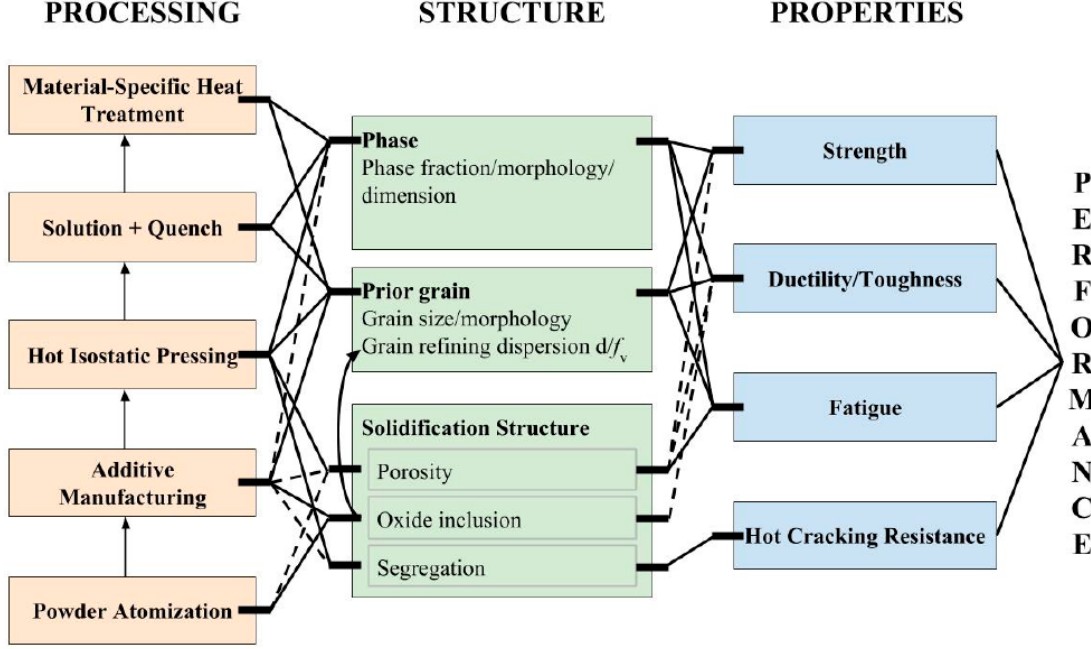

**Figure 1.** Material design chart that showcase the process–structure–properties–performance relationship for additive manufacturing (AM) metal alloy, provided by [13].

Reviews of the AM process have been compiled thoroughly by many authors [14–19]. Discussion on the impact of AM technology on different sectors have been brought up extensively by Ngo et al. [14], whereas author such as Saboori et al. decided to focus on the impact of AM titanium components on the manufacturing process. This section will focus on the seven technologies that were established by the ASTM community, and the benefits of each individual technology to give a broad overview

of the current AM landscape. Table 1 articulates the seven processes that were brought up by the various researchers.

**Table 1.** Description of the AM technologies and benefits associated with it [14,20–23].

| Technologies | Description | Benefits |
|---|---|---|
| Powder-Bed Fusion | Using a laser or electron beam to fuse thin layers of fine powders together, which are spread and closely packed on a platform. Subsequent layers of powders are applied on top of the previous layers until the final part is built | • Fine resolution High quality |
| Direct Energy Deposition | A nozzle mounted on a multi axis arm, which deposits melted material onto the substrate | • Suitable for reparation works <br> • Good mechanical properties |
| Material Jetting | Droplets of material are deposited from the nozzle onto the platform, where it solidifies and subsequent layers are built on it | • Smooth surface finishing <br> • Multi-material printing |
| Binder Jetting | Utilize a binder that was deposited using an inkjet-print head to join materials in a powder bed | • Parts can be made with a range of different colours |
| Material Extrusion | Continuous filament of a polymer is heated and extruded onto the platform or on top of previous layers | • Low cost <br> • High speed |
| Vat Photo-polymerization | A pre-deposited photopolymer in a vat is selectively cured by light | • Fine resolution <br> • High quality |
| Sheet Lamination | Layer-by-layer cutting and lamination of sheets or ribbons of metal | • Low cost <br> • High speed |

## 2.2. Industrial Value

In the past 10 years, many companies have embraced AM technologies and are beginning to enjoy real business benefits from them. In a report by Statista, the global 3D printer market size reached US$7.3 billion in 2017 and the aerospace and defence sectors account for 17.8% of the market distribution in 2016 [24]. The global AM market is expected to see double digit growth into 2022 with market analysis projecting a growth of up to 35% per annum [25]. Recent developments such as cheaper metal powder [26] and the influx of new vendors [27] have significantly reduced the cost of the printers and AM has worked its way into a number of markets. The growing consensus of adopting AM into a production floor is attributed to several advantages over traditional manufacturing, as shown in Table 2.

**Table 2.** Advantages of AM over traditional manufacturing adapted from [28]. Adapted from [28], with permission from Elsevier, 2020.

| Areas of Application | Advantages |
|---|---|
| Rapid Prototyping | • Reduce time to market by accelerating prototyping<br>• Reduce the cost involved in product development<br>• Making companies more efficient and competitive at innovation |
| Production of Spare Parts | • Reduce repair times<br>• Reduce labour cost<br>• Avoid costly warehousing |
| Small Volume Manufacturing | • Small batches can be produced cost-efficiently<br>• Eliminate the investment in tooling |
| Customized Unique Items | • Eliminate mass customization at low cost<br>• Quick production of exact and customized replacement parts on site<br>• Eliminate penalty for redesign |
| Complex Work Pieces | • Produce complex work pieces at low cost |
| Machine Tool Manufacturing | • Reduce labour cost<br>• Avoid costly warehousing<br>• Enables mass customization at low cost |
| Rapid Manufacturing | • Directly manufacturing finished components<br>• Relatively inexpensive production of small number of parts |
| Component Manufacturing | • Enable customization at low cost<br>• Improve quality<br>• Shorten supply chain<br>• Reduce the cost involved in development<br>• Help eliminate excess parts |
| On-site and On-demand Manufacturing of Replacement Parts | • Eliminate storage and transportation cost<br>• Reduce downtime<br>• Shorten supply chain<br>• Allow product lifecycle leverage |
| Rapid Repair | • Reduction in repair time<br>• Opportunity to modify repaired components to the latest design |

A wide variety of materials can be utilized, but metals are generally popular due to their extensive use in industrial and consumer appliances. Figure 2 illustrates the activity map of selected aerospace companies, with many players focusing their research and development work on AM technology. General Electric (GE) leads the industry in terms of the both the volume and machine capacity, and had printed more than 100,000 parts by 2020. Rolls Royce, MTU Aero Engines, Pratt and Whitney and GKN Aerospace have established their own competencies centre to upskill their AM capabilities [25]. GE Aviation has been particularly successful in implementing AM technology into its product. In 2015, GE announced that the next LEAP engine will have nearly 20 3D-printed fuel nozzles [29], simplifying parts by combining multiple components. Traditionally, the aerospace industry used advanced and costly materials like titanium and nickel alloys, which are difficult to manufacture and create a large amount of waste. For example, Wilson et al. [30] has shown that through the use of AM technology, his team was able to achieve a 45% carbon footprint improvement and a 36% savings in total energy over replacing it with an entirely new blade. In 2019, Rolls Royce produced its first AM low-pressure turbine for the Trent XWB-84 which is expected to result in a component weight reduction of up to 40% as well as generate significant cost savings for the company [31].

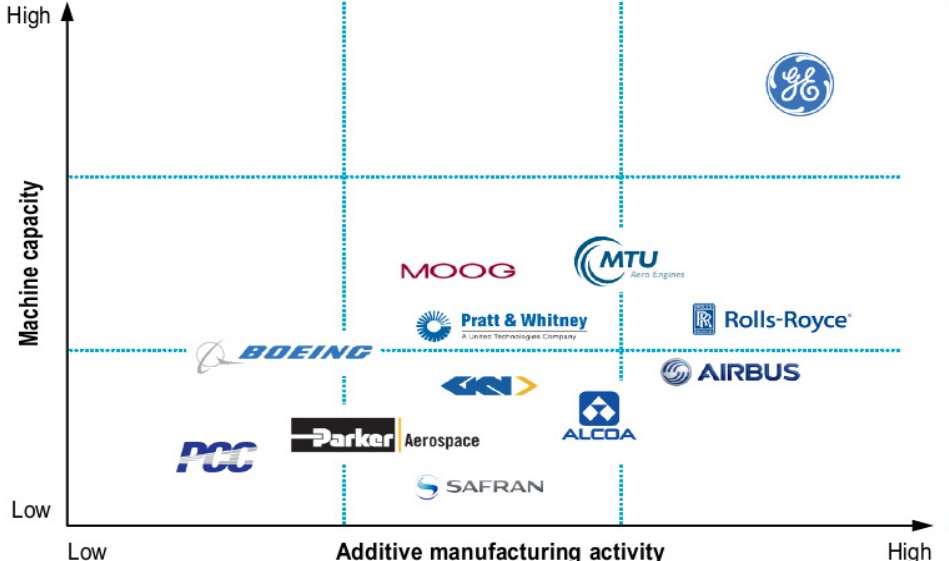

**Figure 2.** AM industry activity map of selected aerospace companies taken from [25], with permission from Roland Berger in 2020.

A separate report by Deloitte highlighted that AM technology has been shown to take the scrap rate to around 10–20% [32] while fabricating parts with intricate geometries such as internal cavities and lattice structures. Moreover, AM has the potential to lower overall cost as it is able to manufacture spare parts on demand, reducing maintenance time and the need for inventory management [33]. Boeing and Airbus typically source 4 million spare parts around the globe, and airlines usually maintain an inventory of spares to avoid their planes from becoming grounded. AM technology is an enabler for these companies to embark on a supply chain transformation, making on-demand manufacturing possible. Combined with the outbreak of COVID-19, companies are looking towards a "just in case" framework rather than "just in time" [34], specifically benefiting from the advanced production capability of AM processes.

### 2.3. Types of Metal AM Process

Out of the seven main AM processes, powder-bed fusion (PBF) and direct energy deposition (DED) are generally used to produce high-quality metal parts. Lewandowski et al. [15] categorized these two mainstream processes into respective energy sources for fusion and the companies that have a specialization in them, as shown in Figure 3.

The DED process has a high degree of control and freedom as it can simultaneously feed multiple types of powders through its nozzle, as shown in Figure 4a. By adjusting the feed rate, it is feasible to achieve desirable microstructural features and chemical composition which is favourable for building functionally graded materials [35] or structural metal components [21]. Apart from manufacturing near net shape components, DED is suitable for repairing high-value parts with little wastage [36–38]. This capability enables manufacturers to remanufacture turbine blades with cracks and voids which is economically viable and helps in design enhancements at the time of restoration [30]. Consequently, remanufacturing using accurate AM processes will enable industries to save energy and material, and contribute towards sustainable design and manufacturing. Despite its benefits, DED faces several challenges in terms of quality and efficiency. Resolutions are generally very low and parts have a rough surface finish that may need post-processing such as by machining to obtain tight tolerances [18].

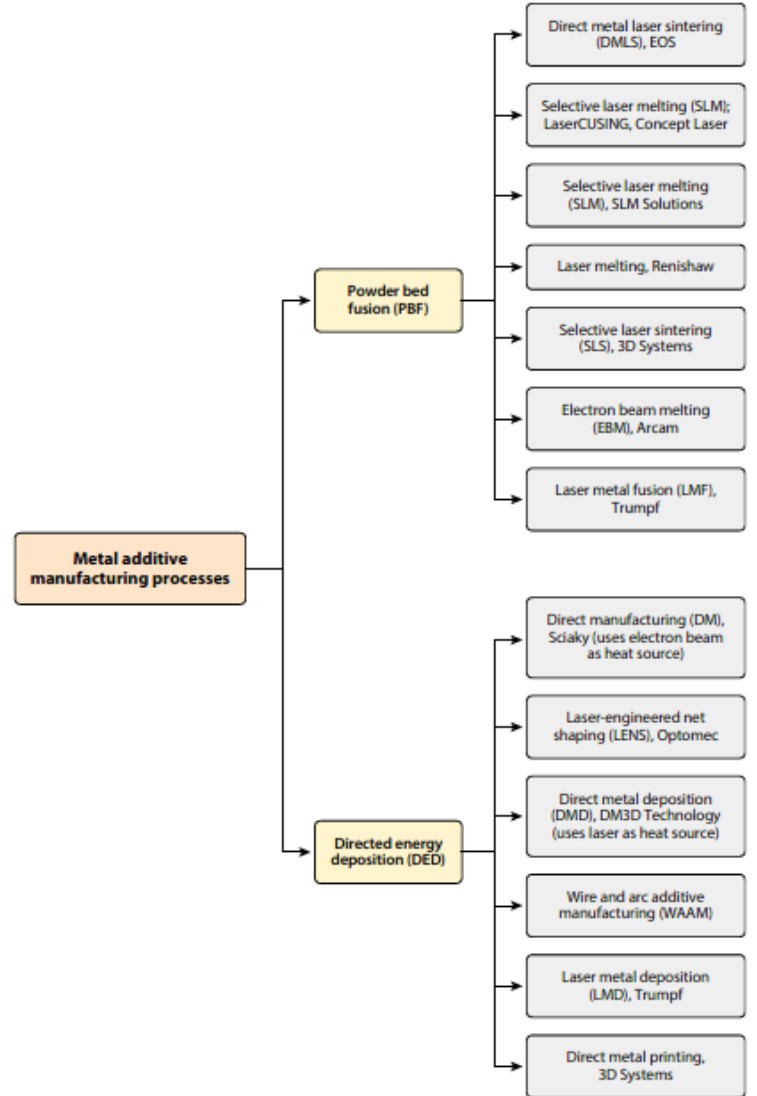

**Figure 3.** Various AM processes adapted from [15], with permission from Annual Reviews in 2020.

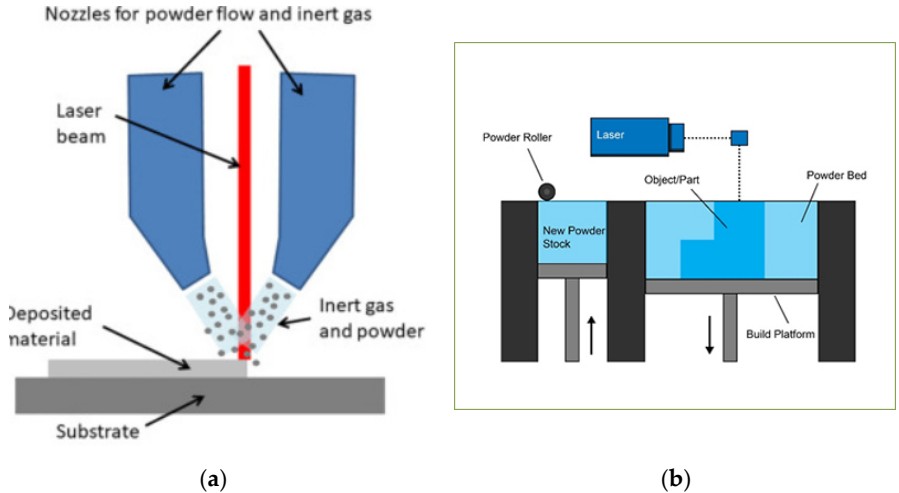

|  |  |
|---|---|
| (**a**) | (**b**) |

**Figure 4.** Schematic diagram of (**a**) direct energy deposition (DED), adapted from [39], with permission from Elsevier, 2020; and (**b**) powder-bed fusion (PBF), adapted from [40], with permission from Elsevier, 2020.

PBF has a unique position because of its potential to manufacture metal components in a range of alloys, at high resolution and accuracy, which has broadened the application to various industries. The process consists of depositing thin layers of fine powder on a platform which is then fused with a laser or electron beam, as shown in Figure 4b. Many metallic materials such as stainless and tool steels, aluminium alloys, titanium and its alloys, and nickel-based alloys can be manufactured by this process.

The main difference between DED and PBF is the way that that powder is fed. In PBF, metal powders are uniformly spread by a rake or roller, while in DED, powders are blown out from the nozzle. The high precision of PBF allows for the optimisation of component design and manufacturing cost. For example, GE aviation has been using metal PBF machines to manufacture its fuel nozzles and next-generation materials, including heat-resistant ceramic matrix composites (CMCs) and carbon-fibre blades. The fuel nozzles were five times more durable than the previous model and reduced the number of required parts from 25 to just five [29].

Another successful case study came from Arconic, which managed to install its 3D printed titanium brackets on the airframe of an Airbus A350 XWB commercial plane, which helps to lower the wastage of raw materials by 80% as compared to manufacturing conventionally [41]. Some of the common materials which have been processed by PBF by other authors are listed in Table 3.

**Table 3.** Common alloys processed by PBF.

| Alloy | Examples | Reference |
|---|---|---|
| Titanium | Ti-6Al-4V, Ti-6.5Al-1Mo-1V-2Zr, Ti-6.5Al-3.5Mo-1.5Zr-0.3Si, Ti-5Al-4Mo-2Zr-2Sn-4Cr, Ti-3Al-10V-2Fe | [42–54] |
| Intermetallics | NiTi | [55–58] |
| Steel | 316L, 17-4PH, AISI 420 | [59–64] |
| Nickel | IN718, IN625, C263, Hastelloy X, K418 | [5,65–100] |
| Aluminium | Al-Si10-Mg, Al-Si12-Mg, 6061 | [49,101,102] |

IN718 is the most commonly used nickel-based alloy in the aerospace industry due to its superior mechanical properties at elevated temperatures and it has been widely used in the turbine section of the aeroengine [103–105]. It has the ability to withstand loading at an operating temperature close to its melting point of 1336 °C [106]. It has a high phase stability of face-centred-cubic (FCC) nickel matrix and the capability to be strengthened by other alloys such as chromium and/or aluminium [107]. The microstructure of IN718 is referred to by Y (gamma), a continuous matrix phase where cobalt and chromium prefers to reside; Y' (gamma prime), an intermetallic phase based on $Ni_3(Al,Ti)$ with a $L_{12}$ crystal structure; Y'' (gamma double prime), a metastable phase that is the primary strengthening precipitate with a body-centred tetragonal (BCT) ordered compound with a $D0_{22}$ crystal structure; δ (delta), an equilibrium phase with an orthorhombic $D0_a$ structure; Laves phase with an embrittling TCP phase; carbides and borides that prefer to reside on the grain boundaries [20,108,109]. However, the usage of PBF IN718 in the aeroengine has been an obstacle owing to the presence of undesirable phases [110] and its unconventional microstructure [80,110,111]. Efforts have been made to limit these defects through the use of heat treatment [5,110] and hot isostatic pressing (HIP-ing) [74], but the results have been mixed and no significant improvements have been made on PBF IN718. Special attention is paid to the microstructure effect of AM IN718 on its material performance.

## 3. Microstructure of AM IN718

### 3.1. Grain Structure

In this review, grain structure constitutes both grain size and grain texture of the material. Unlike its wrought counterpart, AM IN718 display a mixture of columnar and equiaxed grains when no additional treatment is applied. This is due to its uneven cooling rate as the material is being built up layer by layer. Factors such as heat flux and thermal gradients greatly affect the growth of the grains, which are not discussed in this paper. Interested readers could look at the references given

here [93,112–114]. Ahmad et al. [110] showed that AM IN718 has columnar grain growing parallel to the building direction. A magnified image of the microstructure of AM IN718 without any additional treatment using a scanning electron microscope (SEM) is shown in Figure 5.

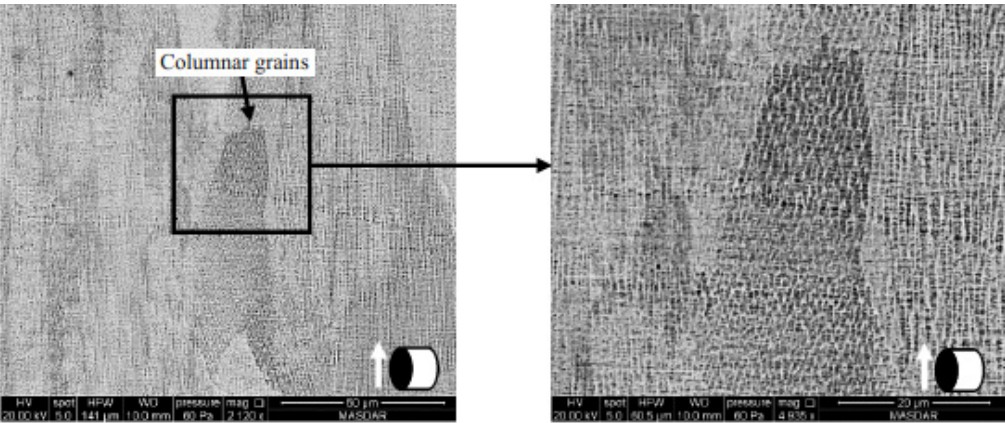

**Figure 5.** Scanning electron microscope (SEM) image showing the columnar grains of AM IN718, taken from [110]

Gribbin et al. [115] took one step further and utilized electron backscatter diffraction (EBSD) to investigate the crystallographic structure of the material. As-built AM IN718 exhibits elongated grain structure with a moderate <100> fibre texture formed along the build direction, as shown in Figure 6. Fatigue strength of the wrought alloy outperforms the AM alloy at room temperature, suggesting the grain texture is likely the main competing microstructural feature affecting the fatigue performances at room temperature. The microstructural study findings were comparable to other studies as well [70,80,86,116], although fatigue life is known generally to be dominated by surface characteristic such as surface roughness [50,117–119] and porosity [79,120], and thus the conclusion made by Gribbin et al. might be incomplete and further investigation has to be undertaken. The fatigue response of the as-built AM material had a similar response to the wrought material at elevated temperature of 500 °C. Both materials had a fatigue limit of approximately 600 MPa [115] despite AM IN718's inherent weakness of high content of δ precipitates, which is known to deteriorate the fatigue behaviour at high temperature. This suggests that the difference in microstructural features is not pronounced in high-temperature environments as compared to the room temperature condition.

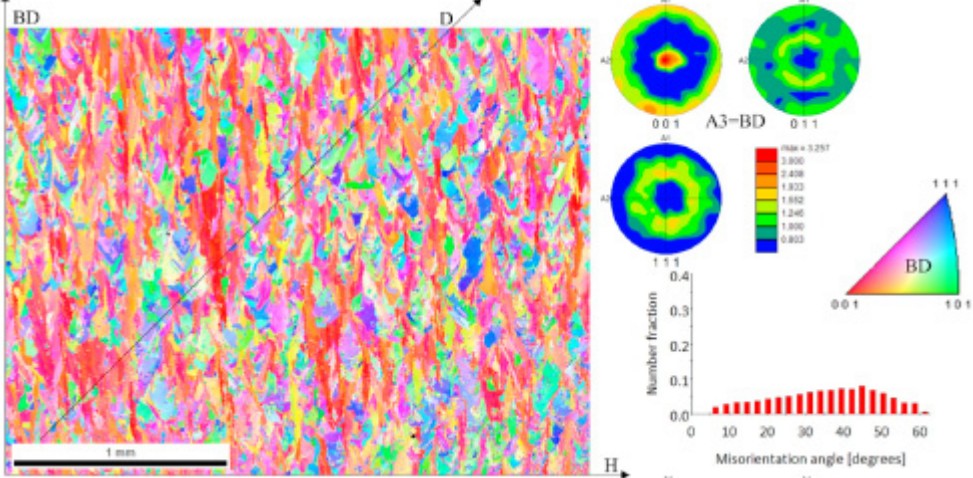

**Figure 6.** Electron backscatter diffraction (EBSD) map and the pole figures showing the crystallographic texture of as-built AM IN718. Adapted from [115], with permission from Elsevier, 2020.

Another interesting finding on wrought IN718 in elevated temperature has [121] shown that the coarse-grain alloy has a fatigue strength significantly lower than fine-grain alloy when it is beyond $10^5$ cycles. It is likely that in order to maximize AM IN718 capability in high-temperature applications, controlling the grain size of the alloy will be vital, and any grain size refinement technique for AM metal alloys will be welcome.

## 3.2. Effects of δ Phases

IN718 is a precipitation-strengthened nickel-based superalloy with Y″ as the main phase contributing to its excellent high temperature strength [122]. However, the metastable Y″ phase easily transforms to a stable δ phase under certain thermal conditions, decreasing the volume fraction of Y′, which indirectly affects the mechanical properties of the alloy [123]. It is generally undesirable as it is known to decrease the fracture toughness and ductility of the material [124,125]. δ precipitates usually formed during the heat treatment process or during service and mainly reside at the grain boundaries [126]. However, there are cases where δ precipitates have been shown to display beneficial effects such as grain stabilization [127] and increasing stress rupture resistance [128]. An et al. [129] investigated the role of the δ phase for fatigue crack propagation behaviour in wrought IN718 and showed that the growth rate increases with increasing δ phase volume fractions. There were both long needle-like and granular shaped δ precipitates present in the alloy which have very different effects on the fatigue crack growth. When Y″ transforms into long needle-like δ precipitates, a precipitates free zone formed around the δ phase, inhibiting micro cracks that are detrimental to the fatigue performance of the alloy. The granular-shaped δ precipitates, with low length-diameter ratio, act as a pin between the grain boundaries, increasing the strength of the alloy.

AM IN718 usually has a slight variation on the volume fraction of its respective phases. In Gribbin's study, the δ phase content in wrought IN718 was 1.6% ± 0.5% while in the as-built AM IN718 it was 3.8% ± 0.4% [115], which is rather unusual for the IN718 alloy. The increase of the δ phase content could be due to the heat treatment used to solution treat the alloy, leaving the precipitates undissolved. Yang et al. [130] compared the microstructure and mechanical performances of PBF-fabricated IN 718 alloy in various heat treatment conditions. The results show that the morphology and distributions of the δ phase are key factors determining their high temperature performance. Too much δ phase along the grain boundaries could cause dislocation to pile up [86], causing local stress concentrations and premature failure. The lack of the δ phase will reduce the strength of the alloy at elevated temperature as it will have limited influence of the pinning effect on grain boundaries.

The formation of intragranular δ precipitates was also observed in AM IN718, which is a common observation for IN718 alloy when the parameters of the heat treatment are not optimized [123]. The presence of a high concentrations of niobium in the feedstock [115], combined with the inconsistent heat flux caused by heating and melting of the powder, is the reason why intragranular δ precipitates are formed. Maximizing the volume fraction of intergranular δ precipitates gives the alloy better ductility while a high amount of intragranular δ precipitates hardens the material [123]. The ratio between intragranular and intergranular precipitates could be a critical parameter in optimizing the mechanical properties at elevated temperature of AM IN718 based on past studies.

## 3.3. Effects of Laves Phases

Niobium is one of the elements present in IN718 and it is highly prone to segregation and tends to form some undesirable phases such as the δ and the Laves phases, which is known for degrading tensile ductility, fatigue and creep rupture properties [131–133]. A high concentration of niobium has been reported by other researchers which catalyse the formation of the Laves phase, depleting the strengthening Y″ phase. The Laves phase provides crack initiation and propagation sites during the melting of the metal powder [132] and is a general observation when IN718 alloy undergoes a process in a high-temperature environment [134] such as heat treatment [5,135,136] or during the powder deposition of the AM process [73,116,137].

The presence of the Laves phase generally deteriorates the ductility, ultimate tensile strength [138] and fatigue life of IN718 alloy [139]. Sui et al. [78] reported that AM IN718 alloy outperforms its wrought counterpart at low stress amplitude due to the role that the Laves phase played during the crack propagation stage. At high stress amplitude, almost all the Laves phase splintered into small fragments that caused microscopic holes or cracks to form. The S-N diagram developed by Sui et al. is shown in Figure 7. The existence of a micron-scaled Laves phase led to local stress concentrations more easily than the wrought alloy, causing it to break up at high stress amplitude. A schematic diagram on the fragmentation of the Laves phase is shown in Figure 8.

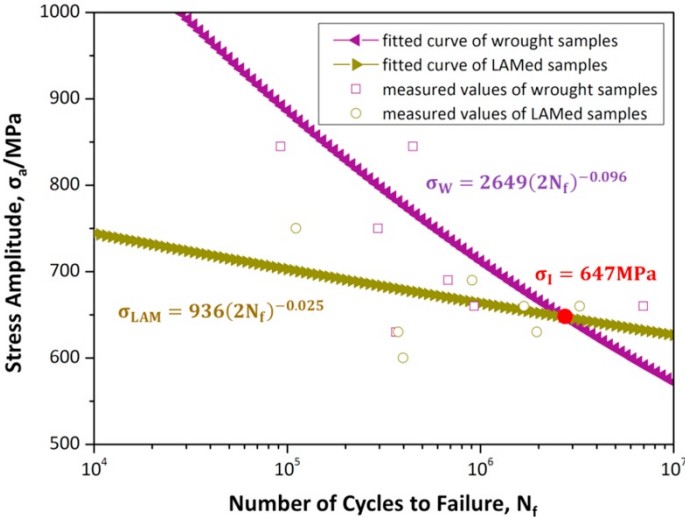

**Figure 7.** High cycle properties of wrought (in purple) and AM (in green) IN718 alloy. Adapted from [78], with permission from Elsevier, 2020.

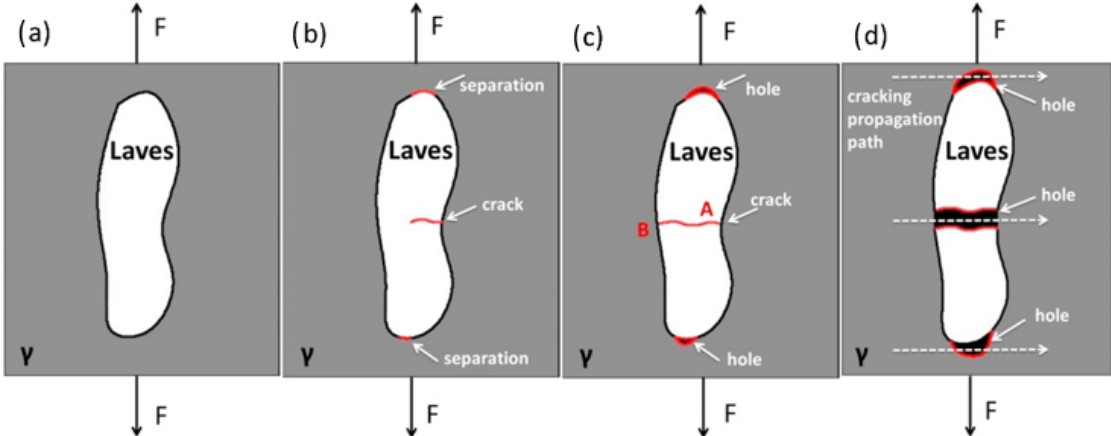

**Figure 8.** Schematic diagram of (**a**) Laves phase present in the matrix, (**b**) crack initiated in the Laves when a tensile force is applied, (**c**) crack propagates through the Laves, (**d**) breaking up of Laves phase at high amplitude. Adapted from [78], with permission from Elsevier, 2020.

To the best of the authors' knowledge, there have been no studies on the effect of the Laves phase on the properties affecting the fatigue life of AM IN718 at high temperature. It is generally regarded as a "parasitic" phase that consumes the available niobium content. Studies have shown that either increasing the homogenization temperature or time [122,140] could lower the volume fraction of the Laves phase. Laves and carbide are generally dissolved resulting in the release of a considerable amount of niobium back to the Y' matrix [136]. With that in mind, it is advisable to formulate new heat treatment standards cater for AM IN718 alloy that is entirely different from the Aerospace Material Specifications (AMS) standards that industry currently uses.

## 4. Mechanical Properties of Powder-Bed Fusion (PBF) IN718 Alloy

Table 4 presents previous work on the mechanical testing of PBF IN718 alloy. Most of the test data are rather similar with slight differences due to either the geometry of the test piece or the direction of the test piece when it is being tested. Generally, a post-processing step such as heat treatment or HIP-ing gives a better tensile strength but with a slight dip in ductility. There were some instances where the tensile properties were superior to the wrought ones, giving the manufacturer extra confidence in employing AM IN718 on its production line. Researchers such as Strößner et al. [111] and Gallmeyer et al. [141] have attempted to optimize the heat treatment process by increasing its homogenization or ageing temperature and thereby controlling the growth of the Y″ phase, and minimizing the impact of either the δ or Laves phases, resulting in an increase in the strength and hardness of the material.

**Table 4.** Summary of mechanical properties of PBF IN718 alloy.

| Condition | UTS/MPa | YS/MPa | El/% | Stress Ratio | Loading Frequency/Hz | Cycles to Failure | Reference |
|---|---|---|---|---|---|---|---|
| As-built | 1110 ± 11 | 711 ± 14 | 24.5 ± 1.1 | - | - | - | [84] |
|  | 1167 ± 10 | 858 ± 12 | 21.5 ± 1.3 | - | - | - | [84] |
|  | 845 | 580 | 21.5 ± 1.3 | - | - | - | [84] |
|  | 1010 ± 10 | 737 ± 4 | 20 | - | - | - | [84] |
|  | 997.8 | 800 | 20.6 ± 2.1 | - | - | - | [84] |
|  | 1335 | 760 | 21.3 | - | - | - | [141] |
|  | 1142.5 ± 5.5 | 898 ± 9 | 22.55 ± 3.35 |  |  |  | [73] |
| As-built, heat treated | 1451 | 1174 | 13.5 | - | - | - | [69] |
|  | 1370 ± 25 | - | 22.2 ± 2 | - | - | - | [77] |
|  | 1221 | 1007 | 16.0 | - | - | - | [82] |
|  | - | - | - | 0 | 20 | $2 \times 10^6$ (run out) | [144] |
|  | 1085 ± 11 | 816 ± 24 | 19.1 ± 0.7 | - | - | - | [111] |
|  | 1010 ± 10 | 737 ± 4 | 20.6 ± 2.1 | - | - | - | [111] |
|  | 1417 ± 4 | 1222 ± 26 | 15.9 ± 1.0 | - | - | - | [111] |
|  | 1387 ± 12 | 1186 ± 23 | 17.4 ± 0.4 | - | - | - | [111] |
|  | 1325 | 620 | 28.6 |  |  |  | SA980 [141] |
|  | 1530 | 1135 | 10.6 |  |  |  | SHT-1 [141] |
|  | 1560 | 1240 | 11.6 |  |  |  | SHT-2 [141] |
|  | 1500 | 1120 | 14.5 |  |  |  | DA620 [141] |
|  | 1580 | 1300 | 9.6 |  |  |  | DA720 [141] |
|  | 1640 | 1245 | 16.6 |  |  |  | SA1020 + A720 [141] |
|  | 1319 ± 39 | 1131.5 ± 29.5 | 16 ± 6 |  |  |  | [73] |
| As-built, hot isostatic pressing (HIP), heat treated | 1200 | 890 | 28 | - | - | - | [109] |
|  | 1384 ± 8 | 1123 ± 13 | 21.5 ± 3.5 | - | - | - | [145] |
| Wrought | 1241 | 1034 | 10 | - | - | - | AMS 5662 [146] |
|  | 1610 | 1160 | 13.5 |  |  |  | [141] |
| As-cast | 802 | 758 | 5 | - | - | - | AMS 5383 [146] |

Data about the fatigue strength of PBF IN718 alloy was limited as it usually costs a significant amount of resources to develop. Fatigue tests are typically conducted on servo hydraulic test machines which are capable of applying large amplitude cycles over a long period of time [142]. They are heavily used in high-value industries such as the aerospace and biomedical sectors where safety standards are much more stringent than in other sectors. For AM IN718 alloy to be used in a safety-critical application, it is vital to understand the process-structure-property relationship, and the availability of fatigue data gives extra confidence for manufacturers to utilize this technology. At the same time, several problems such as weak grain texture and detrimental residual stress [43,115,143] have to be dealt with in order to widen the adoption of AM IN718 alloy. This drives a need to introduce novel post-processing methods to improve the quality of AM products, which will be discussed in the next section.

## 5. Suitability of Surface Enhancement Process

Erfan et al. [147] utilize a common approach in cost, risk analysis and management to review the known surface treatment process in order to evaluate the commercial value that these processes bring

to the AM production line. Ranking is undertaken according to the time and money needed to utilize the treatment, as seen in Figure 9b. Examples of material removal mechanical processes are traditional machining and polishing techniques. They were ranked first, due to the wealth of research findings to make the process more effective and the relatively low cost to employ in AM products. Next in line are processes such as chemical etching [148] and electrochemical polishing (ECP) [149] techniques that are used for parts with intricate geometries such as lattice and cellular structure which are hard to reach. In addition, these chemical techniques have been able to decrease surface roughness with an average and max height improvement of 73% and 65% respectively for ECP [149].

**Cost Analysis Matrix**

| Time | | | Money | | | | |
|---|---|---|---|---|---|---|---|
| Very High | 5 | 5 | 10 | 15 | 20 | 25 | |
| High | 4 | 4 | 8 | 12 | 16 | 20 | |
| Moderate | 3 | 3 | 6 | 9 | 12 | 15 | |
| Low | 2 | 2 | 4 | 6 | 8 | 10 | |
| Very Low | 1 | 1 | 2 | 3 | 4 | 5 | |
| | | 1 | 2 | 3 | 4 | 5 | |
| | | Very Low | Low | Moderate | High | Very High | |

| Surface post-treatment | | Cost | | Final Score | Rank |
|---|---|---|---|---|---|
| | | Time | Money | | |
| Material removal | Mechanical | Low | Very Low | 2 | 1 |
| | Laser-based | Very High | Very High | 25 | 5 |
| | Chemical | Low | Low | 4 | 2 |
| No material removal | Mechanical | Moderate | Low | 6 | 3 |
| | Laser-based | Very High | Very High | 25 | 5 |
| Coatings | | High | High | 16 | 4 |
| Hybrid treatments | | Very High | Very High | 25 | 5 |

(a)    (b)

**Figure 9.** Taken from [147], with permission from Elsevier, 2020. (**a**) cost analysis matrix (**b**) scores and rank of various surface treatment process in terms of cost analysis.

On the other hand, surface treatments such as shot-peening [150], deep-rolling [151] and laser shock peening (LSP) [152] are categorized under no material removal mechanical treatments and are commonly used to increase the usability of AM materials. For high-temperature applications, LSP has additional advantages over other surface treatments due to its ability to impart deep compressive residual stresses [153], lower cold work on the surface [154] and the ability for grain refinements [155,156]. Inherently, the LSP process introduces high strain rates up to $10^6$ s$^{-1}$ which generate beneficial dislocations near the surface layer [157–159].

Table 5 compares the effects of shot peening and LSP on strain rate, cold work, depth of influence and the typical roughness of various metal alloys. Generally, LSP brings about a significant lower cold work as compared to shot peening. A lower amount of cold work was observed in Ti-6Al-4V [160] and IN718 coupons where the cold work was between 1–6% [154]. The amount of cold work that is being induced by LSP boils down to the material's deformation capability. Although it is beyond the scope of this work to discuss shot peening, it is useful to bring it up as a comparison to LSP as it is a widely adapted technique in the industry. Shot peening involves multiple steel or ceramic shots projected at a high velocity through a nozzle, striking the surface with force sufficient to create plastic deformation. Due to the continuous shots, most of the energy is expended in inducing plastic deformation, resulting in a highly cold-worked surface layer [160]. On the other hand, LSP produces remarkably low cold work on the surface at room temperature [161], as shown in Figure 10. The cold work produced by shot peening comes close to 0% after a depth of $10 \times 10^{-3}$ inch, dropping from the initial 30%. This begs the question of why LSP is able to drive similar or higher compressive residual stresses and yet produce a lower amount of cold work on the surface.

**Table 5.** Comparison of shot peening and laser shock peening (LSP).

| | Strain Rate (s$^{-1}$) | Cold Work (%) | Depth (mm) | Typical Roughness *Ra* (µm) |
|---|---|---|---|---|
| Shot peen (X20 Steel) | $10^3$–$10^4$ | 15–50 | 0.2 | 4.52 |
| LSP (X20 Steel) | $10^6$–$10^7$ | 5–7 | 1.2 | 0.98 |
| Shot peen (Ti-6Al-4V) | - | 75 | Surface | - |
| LSP (Ti-6Al-4V) | - | 1–2 | Surface | - |
| Shot peen (IN718) | - | 30 | Surface | - |
| LSP (IN718) | - | 3–6 | Surface | - |

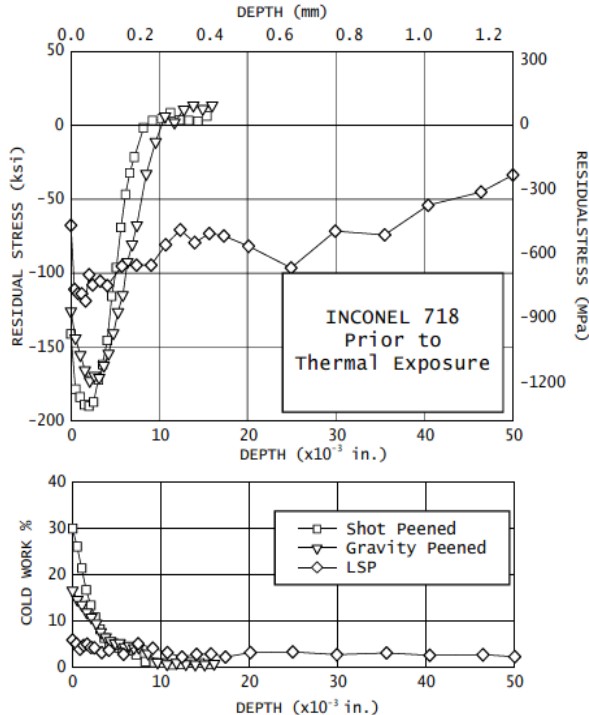

**Figure 10.** Taken from [160], Residual stress distribution and cold work developed by in IN718 coupons using various surface enhancement process.

A high degree of cold work has been found to relax rapidly at high temperature [160–162] which is detrimental for high temperature application in the nuclear and aerospace industries. LSP might be in a more advantageous position that shot peening if it is able to withstand thermal stress relaxation in these industries. More investigation has to be done to find out the process-structure-property-performance relationship, utilizing advanced material characterization methods such as EBSD or transmission electron microscopy (TEM).

There are interesting initiatives where LSP is being used as a post-processing step for AM metal components such as aluminium [163], stainless steel [164] and titanium alloy [152]. Kalentics et al. [164,165] have proposed using LSP to tailor the residual stresses of stainless steel samples by moving the baseplate back and forth from a printing machine to an LSP station. He has dubbed it as 3D LSP, an ex-situ LSP and AM process which has been shown to increase both the magnitude and depth of compressive residual stress. The depth of compressive residual stress could reach up to 1 mm for an AM 316L stainless steel component subjected to the 3D LSP principle, as shown in Figure 11. This research is a promising start to combine LSP techniques into AM material, making it more useable for aerospace applications.

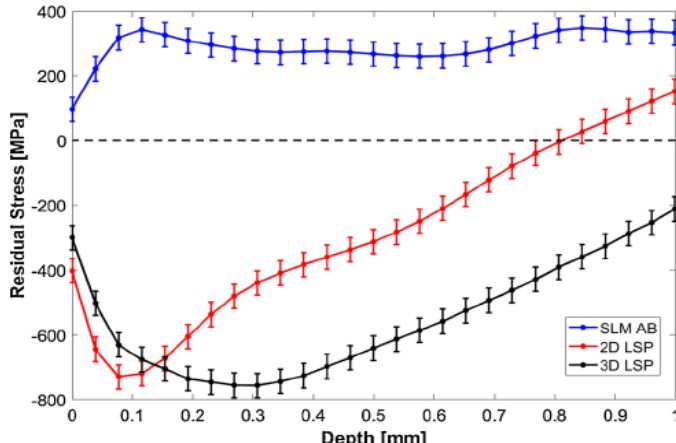

**Figure 11.** Taken from [164], with permission from Elsevier, 2020. Residual stress curve measured for 316L Stainless Steel for as-built (AB), LSP after AM (2D LSP) and ex-situ LSP on AM (3D LSP). Adapted from [164], with permission from Elsevier, 2020

## 6. Conclusions

The current priority of aeroengine manufacturers is to investigate the applicability of AM components in their manufacturing process as it offers significant processing flexibility and potential cost reduction. IN718 is one of those metal alloys that is suitable for the AM process route as it allows manufacturers to process it in an easy and straightforward way that ensures that the material properties are still well preserved. With layer-wise building of components, the process produces up to about 5% waste, reducing the raw material wastage significantly as compared to manufacturing it conventionally for aeroengine components.

Inherently, as-built AM products give a mixture of columnar and equiaxed grains which directly impacts the mechanical properties of the material at room temperature. At high temperature usage, its effect is not as prominent, and grain size will be the crucial factor in determining its fatigue life. Further research should be performed to identify the effects of the various phases present in the alloy that affect the usability of the material. It should be possible to control the growth of $\Upsilon''$, $\delta$ and Laves to maximize the properties of AM IN718 alloy as suited for its intended application.

Numerous heat-treatment procedures have been attempted by the scientific community to match its tensile strength with the wrought alloy. Higher homogenization temperature and longer ageing time is usually employed to reduce the presence of harmful phases such as the $\delta$ and Laves phases. Such knowledge is important for developing beneficial microstructures and material properties for the intended application.

Analysis obtained from the literature suggest that the mechanical properties of PBF IN718 alloy were very similar to its wrought counterpart. However, there were insufficient experimental data to showcase the fatigue lifespan of AM IN718 alloy due to the complexity and high cost of the experiments.

Surface enhancement techniques were explored in this study as this could assist AM IN718 alloy to perform better in high-temperature applications. LSP has the potential to be a suitable technique as it can induce a lower amount of cold work which is beneficial in a high-temperature environment.

**Author Contributions:** Conceptualisation, literature review and writing of the manuscript were done by C.K.Y. Supervision and review of the manuscript performed by G.J.G., G.W. and C.C.W. All authors have read and agreed to the published version of the manuscript.

**Funding:** This research was funded by Agency for Science, Technology and Research (A*STAR) Singapore and Warwick Manufacturing Group (WMG), University of Warwick.

**Acknowledgments:** This research is part of the EngD project that is sponsored by A*STAR and the authors are thankful for the funding.

**Conflicts of Interest:** The authors declare no conflict of interest.

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
