# Peer review of "A Critical Review of the Material Characteristics of Additive Manufactured IN718 for High-Temperature Application"

_metals, doi:10.3390/met10121576_

Round 1
Reviewer 1 Report
The paper provides a good review on AM IN 718
Some comments below
.1. Introduction usually does not have figures.
2. You cannot use VAT lamination and other techniques for 718 so why mention them. It makes no sense
3. Line 141
4. Surface enhancement processes such as electroplating are discussed by several autors . see Fayazfar et al. (Vacuum). These discussions should also be included.
5. Effect of printed orientation should also be discussed. Look at Kesharawarzkermani and Esmaielizadeh et al.
6. The authros have added a few details on 718 but it is suggested to increase the discussion. A lot of work needs to be done in this regard to make it a good review paper.
Author Response
Thank you for your time in reviewing the paper. I have made changes to the manuscript accordingly to some of the points you have raised.
- Yes, I have shifted them to the next section so that it looks better
- I have decided to include these techniques to give an overview of the seven AM processes.
- Thank you for pointing it out. I have edited it accordingly
- Yes, I have added in ECP for discussion. I have decided to focus on surface enhancement process as this area is rarely discussed
- I have decided to opt out the discussion on the effects of building orientation on AM component as the difference in the performance can be relatively small. I have added in a few sentence to clarify that I would not be discussing this point due to the fact that there are many reviews have been done on this topic
- I will make improvements on it. Thank you
Reviewer 2 Report
The manuscript „A Critical Review of the Material Characteristics of Additive Manufactured IN718 for High Temperature Application“ did not inspire me at all. First of all the word „Critical“ seems to me not to be relevant, because one would expect a detailed comparison of microstructure and mechanical properties of AM IN718 with wrought and cast IN718. Even more embarrassment makes using of “for High Temperature Application” in the title and abstract, because there is absolutely nothing about high temperature properties in the manuscript. In contrary a significant part of the manuscript is devoted to the AM methods, which is not reflected in either the title or the abstract
I can respect a lot of work hidden behind the manuscript and that the paper could be interesting for several readers. That is why I am recommending to invent a new title and to rewrite accordingly the abstract to hinder that the paper does not meet the expectations of potential readers.
Author Response
Thank you for your time in reviewing this paper. I have made changes to some of the feedback that you have given.
I would like to take this chance to clarify the purpose of this review paper.
- Research work on wrought and cast IN718 is extensive and giving a detailed comparison would be too broad. I have decided to zoom in on the differences that AM has bring (e.g. textured microstructure and precipitates formation)
- High temperature application have been mentioned in this paper (Line 146, 184, 219, 236, 303). There are many aspect of high temperature application (e.g. creep, corrosion due to oxidation and thermomechanical effect). Due to the nature of these experiments, it is costly to perform this type of research and results generated by these type of experiments remain very elusive. Thus, I have decided to assess through the material aspect of it (e.g. texture, precipitates, cold work)
- I hope you understand that this review paper is to give a broad overview of AM and how IN718 will be suitable for the AM process route. It is natural to take a look at other enhancement processes to bring up the useability of AM IN718 in a high-temperature environment, which is predominantly used in the aerospace, marine and energy sector.
Reviewer 3 Report
Dear Authors,
I like your article since the review articles are valuable in my opinion in most cases. However, I have found some issues to discuss/correct. Please follow my remarks and after you address them I do recommend the article for further processing.
- Line 61 – citations [6-11] – please extend it writing 1-2 sentences explaining what has been stated in each one.
- Line 64 – the same issue and I wonder perhaps it would be good to include some precise quotations in the Table 1?
- Line 77 – some problem with the reference has been displayed here.
- Line 141 – the same problem as above
- Table 3 – again, the references as [57-93] are not very helpful.
- Figure 6 – low quality
Sincerely,
Reviewer
Author Response
Thank you for your time in reviewing my paper. I have edited the paper accordingly to some of your points. Kindly see my response to each points that you have raised.
- Yes, I have added in some comments on the citation to give readers some direction on what to look out for.
- Table 1 is a condensed version of the description of various AM techniques described by the literature. I have listed out the benefits of each techniques by consolidating all the ideas given by the literature. It will be difficult to give some precise quotation as it might swamp the readers with too much information
- Yes, I noticed that when I submited the manuscript through the system. Thank you for pointing out and I have made changes to it
- Similar to point 3
- Table 3 is an overview of what alloy that have been used by LPBF AM technique. My intention here is to show that these material are the choice of material for LBPF process.
- I apologise for the low quality of the figure. This is provided by the original author itself